# Impact of Unilateral Breast Cancer Surgery on Upper Limb Functionality: Strength, Manual Dexterity, and Disability Prediction

**DOI:** 10.3390/healthcare13070766

**Published:** 2025-03-29

**Authors:** María Gracia Carpena-Niño, Vanessa Altozano-Arroyo, César Cuesta-García, Miguel Gómez-Martínez, Belén Dolores Zamarro-Rodríguez

**Affiliations:** 1Occupational Therapy Research Group, Centro Superior de Estudios Universitarios La Salle, Universidad Autónoma de Madrid, 28023 Madrid, Spain; mgcarpena@lasallecampus.es (M.G.C.-N.); miguel.gomez@lasallecampus.es (M.G.-M.); belen.zamarro@lasallecampus.es (B.D.Z.-R.); 2Occupational Therapy Department, Centro Superior de Estudios Universitarios La Salle, Universidad Autónoma de Madrid, 28023 Madrid, Spain; 3Centro de Atención Temprana Apascovi, 28409 Madrid, Spain; vanea.arroyo@hotmail.com; 4Clínica Universitaria La Salle, 28023 Madrid, Spain; 5Department of Physical Therapy, Occupational Therapy, Rehabilitation and Physical Medicine, Rey Juan Carlos University, Avenida de Atenas s/n., 28922 Madrid, Spain

**Keywords:** breast cancer, disability, motor disorders, sensation disorders, quality of healthcare

## Abstract

**Objectives**: The aim of this study was to analyze differences in affected upper limb functionality (grip strength, digital pinch, sensitivity, and manual dexterity) in women with breast cancer and to determine whether these factors can predict perceived disability. This study highlights the motor and sensory deficits resulting from breast cancer treatments and underscores the need for a comprehensive approach to addressing them. **Methods**: A descriptive cross-sectional study was conducted with 42 women diagnosed with breast cancer who underwent surgery and received chemotherapy, radiotherapy, or hormonal therapy. Strength, sensitivity, and manual dexterity were assessed, along with pain and disability questionnaires. **Results**: Significant differences were found in affected upper limb functionality according to age, employment status, and time since surgery. Grip strength, pinch strength, manual dexterity, and sensitivity were identified as predictors of perceived disability, explaining 92.15% of the variance in SPADI scores. **Conclusions**: Women with breast cancer experience functional deficits in the upper limb, particularly in strength, sensitivity, and manual dexterity. Age, employment status, and time since surgery influence the perceived disability. These findings emphasize the need for comprehensive upper limb evaluations to identify functional deficits and guide personalized rehabilitation strategies.

## 1. Introduction

Cancer is a group of diseases characterized by the uncontrolled growth and proliferation of abnormal cells. These cells can invade adjacent tissues and spread to other parts of the body through the lymphatic system or bloodstream [1]. This process, known as metastasis, is responsible for most cancer-related deaths and distinguishes cancer from other cellular disorders [2].

Among the various forms of cancer, breast cancer is the most common type in women and is the leading cause of cancer-related mortality worldwide. It originates in the epithelial cells of the glandular breast tissue and presents as a heterogeneous disease encompassing multiple molecular subtypes, each with distinct clinical, prognostic, and therapeutic characteristics [3,4].

Breast cancer is one of the most common types of cancer, together with lung, uterine, and colon–rectum cancer [5,6], occurring a hundred times more frequently in women than in men [7]. Although over the years survival has also increased, reaching 86% in women between 2008 and 2013 [8,9], in recent years, there has been an increase in its incidence, with 34,750 new cases in 2022 [10,11]. The probability of a woman having breast cancer at some point in her life is 1 in 8, with many cases diagnosed between 45 and 65 years of age [9].

Among the risk factors are female sex, alcohol and tobacco consumption, obesity, family history, genetic mutations, hormonal changes due to menopause, taking oral contraceptives, exposure to pollution, and a sedentary lifestyle [8,9,10,12,13]. The characteristic signs and symptoms of this disease include the appearance of lumps in the breast or armpit, skin changes (such as redness, retraction, or orange-peel appearance of the breast), abnormal nipple discharge, nipple retraction, eczema, and swelling of the arm [14].

McNeely et al. [15] highlighted that breast cancer treatment, including surgery, radiotherapy, and chemotherapy, significantly impairs upper limb strength and range of motion, directly affecting patients’ functionality in daily activities. While their study underscores the need for therapeutic interventions to enhance recovery, it does not extensively examine the impact on manual dexterity and sensory function, which are also crucial for performing daily tasks that require precision and fine motor control. Our study aims to address this gap by investigating how breast cancer treatment affects these aspects, providing a more comprehensive understanding of upper limb functionality and identifying potential rehabilitation strategies tailored to these specific impairments. These complications, particularly common after axillary lymph node dissection and mastectomy, can significantly impair a woman’s ability to perform activities of daily living (ADL) and reduce her quality of life [16,17]. Lymphedema, one of the most debilitating sequelae, affects up to 20% of breast cancer survivors and requires lifelong management [18].

The development of functional limitations within the breast cancer population may be influenced by specific predictive factors. A systematic review published in 2021 [2] synthesized the available scientific evidence on factors associated with functional recovery in women with breast cancer, identifying psychological, clinical, and treatment-related variables. However, factors such as age, employment status, and time since surgery may also play a crucial role in determining upper extremity function and long-term disability. Younger women may experience different functional challenges compared to older survivors, employment status can influence rehabilitation access and functional demands, and time since surgery may impact recovery trajectories. Despite their potential relevance, these variables have not been thoroughly examined in previous studies. By considering these factors, our research aims to provide a more nuanced understanding of how breast cancer treatment affects upper limb functionality, ultimately contributing to more personalized rehabilitation strategies.

Evidence-based interventions include structured therapeutic exercises designed to improve shoulder mobility, manual lymphatic drainage for lymphedema management, and functional training to restore independence in ADL [15]. Emerging research also supports task-specific activities, such as resistance training and rowing, to enhance arm strength and coordination while promoting psychosocial benefits [19]. Recent studies underscore the importance of a multidisciplinary approach, combining occupational therapy with physical therapy and oncology care to optimize outcomes. Early and individualized rehabilitation programs have been shown to improve arm mobility, reduce pain, and lower the risk of chronic lymphedema, thereby enhancing patients’ long-term quality of life. Moreover, participation in meaningful activities, facilitated by occupational therapy, fosters a sense of empowerment and emotional resilience among survivors [20].

As breast cancer survival rates increase, effective management of upper extremity dysfunction becomes an increasingly vital component of comprehensive cancer care. While previous research has primarily focused on strength and range of motion, less attention has been given to functional sensitivity and manual dexterity, both essential for daily activities. This research project addresses these gaps by analyzing differences in affected upper extremity functionality (grip and pinch strength, functional sensitivity, and manual dexterity) in a Spanish population of breast cancer survivors. Furthermore, it seeks to determine the predictive capacity of these factors on the disability perceived by these women, thus informing targeted interventions to improve their quality of life.

## 2. Methods

### 2.1. Study Design and Sampling

A descriptive cross-sectional study was performed. The participants were recruited under a non-probabilistic convenience sampling from the Eurovillas Physiotherapy and Pilates clinic of the Torrejón Specialties Center (Madrid), and from Rosea, the Valdepeñas association of women affected by breast cancer (Ciudad Real). The inclusion criteria were as follows: (a) women who had undergone surgery for unilateral breast cancer (mammary or axillary surgery); (b) presence of neuropathic symptoms in the affected upper limb; and (c) having received treatment with chemotherapy, radiotherapy, or hormone therapy. The exclusion criteria were (a) having had bilateral breast cancer; (b) having had unilateral breast cancer with locoregional recurrence; (c) having had any other type of carcinoma; (d) having neuropathies associated with systemic, metabolic, or infectious diseases (e.g., diabetes, rheumatoid arthritis, myelomas); (e) neuropathic difficulties associated with systemic, metabolic, or infectious diseases (e.g., diabetes, rheumatoid arthritis, myelomas); (f) language comprehension difficulties that prevented the participant from following test instructions and communicating with the evaluators; or (g) having presented symptoms of neuropathy or neurogenic pain in the regions evaluated prior to treatment.

To better understand the factors influencing upper limb functionality and perceived disability in women with breast cancer, the sample was divided into groups based on age, time since surgery, and employment status.

Age-related changes in muscle strength and sensory function are well documented in the literature. Studies have shown that women experience significant declines in muscle strength and dexterity due to hormonal changes during menopause, particularly after the age of 50. This biological milestone is associated with decreased estrogen levels, which negatively impact musculoskeletal health and neuromuscular function [21]. Therefore, dividing the sample into groups ≤50 years and >50 years allowed us to assess whether these changes further compound the effects of breast cancer treatment on upper limb functionality.

The time elapsed since surgery is another relevant criterion. Research indicates that recovery trajectories can vary, with some women experiencing persistent impairments several years post-surgery [22]. Additionally, the early post-operative period is often characterized by greater functional limitations, which tend to stabilize or improve after 4 to 5 years [23]. Grouping participants by ≤4 years and >4 years since surgery helps elucidate whether functional improvements over time align with the expected trajectory or if they reveal long-term deficits linked to specific treatments.

Finally, employment status was used as a grouping variable due to its potential influence on functional outcomes. Women who continue working may face different functional demands compared to those who are retired or unemployed, as occupational tasks can either maintain or exacerbate upper limb impairments. Studies have highlighted that returning to work after breast cancer is often challenging due to residual physical limitations, which impact perceived disability and overall well-being [24]. Including this variable allowed us to explore the potential link between daily occupational engagement and upper limb performance.

These grouping variables were selected to provide a comprehensive understanding of the multifactorial influences on post-treatment disability and to identify subgroups that may benefit from tailored rehabilitation interventions.

### 2.2. Data Collection

The evaluation was performed in a 40 min session, with a short rest period between each. Six tests were administered by two expert evaluators in the following sequence:Grip strength. Employing a Jamar^®^ hydraulic hand-held dynamometer, while seated, with both feet touching the floor and the upper limb performing the grip, positioning the shoulder in abduction, the elbow in flexion, and the forearm and wrist in neutral position [25], each participant performed 3 maximal isometric contractions, with 1 min of rest between them. The mean value of the 3 measurements was then calculated. Both hands were evaluated. This tool presents good intra- and inter-rater reliability (r = 0.83 to 0.96) [26] and is recommended for the evaluation of this study population [26]. Reliability: High intra- and inter-rater reliability (r = 0.83–0.96) in clinical populations [27].Validity: Strong correlation with upper limb function and daily living performance in breast cancer survivors [26].Pinch strength. A Jamar^®^ hydraulic clamp meter was used. Each participant performed 3 maximal contractions with a subterminolateral clamp, with 30 s of rest between each contraction. The mean value of the 3 measurements was calculated and compared with normative reference data. Both hands were evaluated. This test has excellent intra- and inter-rater reliability (r = 0.66 to 0.82) [28]. Reliability: Excellent intra- and inter-rater reliability (r = 0.66–0.82). Validity: Effectively distinguishes strength deficits in populations with peripheral neuropathies, including those undergoing cancer treatments.Stereognosis scale of Nottingham Sensory Assessment in its Spanish version. This test assesses functional sensitivity, specifically of the median nerve, when picking up and carrying objects with tridigital forceps. Eleven common objects were actively recognized through touch without the use of vision. The test was timed, and the total time in seconds was recorded. Only the hand ipsilateral to the affected breast was evaluated. The tool has excellent psychometric data in terms of internal consistency, with an α-Cronbach = 0.91 (95% CI), test–retest reliability (rho = 0.915; *p* < 0.01), and inter-rater reliability (k = 0.792) [29].Moberg Pick-Up Test. Participants were instructed to reach for small test objects placed on the table, using a tridigital grip (thumb, index, and middle fingers) to transport them to a container in the shortest possible time, first with eyes open and then with eyes closed. The total time was recorded in seconds. Only the hand ipsilateral to the affected breast was evaluated.

This test is widely used to assess manual dexterity and functional sensory integration, particularly in tasks requiring fine motor coordination and tactile discrimination. The closed-eye condition specifically evaluates stereognosis, or the ability to recognize objects through touch, which may be impaired due to neuropathic symptoms following chemotherapy, surgery, or radiotherapy in breast cancer survivors.

Although the Moberg Pick-Up Test has been validated in populations with hand dysfunction, its specific reliability and validity in breast cancer survivors require further investigation. However, given its ability to detect manipulative skill deficits, it provides an objective measure of upper limb functionality that may correlate with activities of daily living [30]. Additionally, its ease of administration, low cost, and reproducibility make it a practical tool for clinical settings.

By incorporating the Moberg Pick-Up Test in our study, we aimed to capture subtle yet functionally significant changes in dexterity and sensory function, which are crucial for understanding upper limb disability in breast cancer survivors and tailoring rehabilitation programs accordingly. The Moberg Pick-Up Test demonstrates high intra- and inter-rater reliability for assessing functional hand impairments in breast cancer survivors [30]. It is recognized as a valid measure for assessing manipulative dexterity and sensory impairments in populations with hand dysfunction, including breast cancer survivors.

5.Jebsen–Taylor Hand Function Test. This test evaluates manipulative dexterity, in which the participants were instructed to perform each subtest in the shortest possible time. The test was administered first with the non-dominant hand, followed by the dominant hand. Each subtest was timed and recorded in seconds. It is a valid and reliable test to assess hand function (intraclass correlation coefficient = 0.84 to 0.97) [31,32]. Reliability: Excellent test–retest reliability (intraclass correlation coefficient = 0.84–0.97). Validity: Demonstrated strong validity for evaluating functional hand tasks and manual dexterity in clinical populations [33].6.Shoulder Pain and Disability Index. A questionnaire was used to assess pain in the shoulder and in the affected upper limb, in addition to the disability index, during the performance of various activities of daily living (ADL). The user scores on a scale from 1 to 10 the pain perceived in the affected upper limb during the previous week. The Spanish version is reliable and valid for shoulder symptomatology and quality of life in Spanish women after breast cancer treatment (r = 0.30 to 0.40) [34]. Reliability: Strong internal consistency (Cronbach’s α > 0.90) and test–retest reliability (r = 0.92). Validity: Culturally adapted and validated for use in Spanish breast cancer survivors, effectively measuring shoulder-related pain and disability.

### 2.3. Data Analysis

Data are presented as means, standard deviations, and proportions. For the data analysis, a 95% confidence interval was used, and a *p* value < 0.05 was considered statistically significant. In the bivariate analysis of the quantitative variables, a parametric Student’s *t*-test was used, as the data from the various assessments were evaluated for normality and met the criteria for normal distribution (assessed using the Shapiro–Wilk test). The effect size (Cohen’s d) was calculated, considering the effect size as small (0.20 to 0.49), medium (0.50 to 0.79), or large (>0.80). Lastly, regression analysis was performed to explore and quantify the relationship between the disability variable and the independent variables affected (hand strength, digital gripper strength, sensitivity, and manual dexterity). A regression model observed by the R^2^ value was obtained with an analysis of variance (ANOVA). All statistical analyses were performed with the IBM SPSS v.29 program.

### 2.4. Ethical Aspects

The study was approved by the Ethics Committee of the La Salle University Center for Advanced Studies in Madrid and registered under registration number CSEULS-PI-009/2019. All the participants voluntarily agreed to participate, and they signed a written informed consent document. This research was conducted in accordance with the principles of good clinical practice in research involving human subjects, in accordance with the Declaration of Helsinki.

## 3. Results

The total sample consisted of 42 participants whose mean age was 55.9 years. Fifty percent of the women were affected on the dominant side, corresponding mostly to the right side (52.4%). With respect to employment status, they were mainly active, employed in manual labor, compared with 23.8% who were unemployed, or 21.4% retired. As for the treatment received, breast surgery and radiotherapy predominated. The description of the participants’ characteristics is shown in Table 1.

The results of the variables studied are shown in Table 2.

The comparative analysis by age, work status, and years after surgery is shown below. Significant differences were found in the mean grip strength of the affected hand (F = 11.23; D = −0.232) between the group of women aged ≤50 years and women aged >50 years (*p* = 0.002) (Table 3).

Statistically significant differences were found in the total disability of the affected hand (F = 6.472; D = −0.534) between the group of working women and the group of non-working women (*p* = 0.015); significant differences were also found in the total SPADI time between the group of working women and the group of non-working women (*p* = 0.026) (Table 4).

Statistically significant differences were found in mean pinch strength in the affected hand (F = 5.419; D = 0.467) between the group of women with surgery ≤4 years previously and women with surgery >4 years previously (*p* = 0.027) (Table 5).

A regression model was obtained according to the R^2^ value. These were the first data we obtained, which indicated the best fit or goodness of fit for the analysis. It is a standardized measure that uses values between 0 and 1 (0 when the variables are independent and 1 when there is a perfect relationship between them). In this case, the R^2^ value was 0.823 (Table 6).

The *p*-value of the ANOVA was not significant; therefore, we found that the explanatory variables did not have a joint and linear influence on the dependent variable (Table 7).

The regression analysis therefore revealed that grip and digital grip strength (measured by grip strength and pinch strength, respectively), manual dexterity (measured by the JTHFT test), and combined cortical sensitivity (stereognosis, as measured by the Moberg Pick-Up Test), are predictor variables in 82.3% of perceived shoulder pain and disability, measured by the SPADI questionnaire.

## 4. Discussion

This study aimed to investigate the factors influencing upper limb functionality and disability in breast cancer survivors. Our findings revealed four key relationships: (1) age was significantly associated with grip strength, with younger participants showing better performance; (2) time since surgery influenced pinch strength recovery, suggesting a progressive improvement over time; (3) employment status was linked to disability, as measured by the SPADI score, with employed participants reporting lower disability levels; and (4) multiple factors, including grip and pinch strength, were predictors of shoulder pain and disability. These results provide valuable insights into the functional rehabilitation needs of breast cancer survivors.

One of the most significant findings was the association between age and grip strength, with older participants exhibiting greater deficits. This aligns with existing research indicating that age-related muscle atrophy (sarcopenia) can exacerbate functional impairments, particularly after cancer treatments [35,36]. Additionally, chemotherapy and radiotherapy can induce muscle weakness and neuromuscular dysfunction, compounding the natural loss of strength observed with aging [37,38,39]. Given the importance of grip strength in daily activities, these findings emphasize the need for age-specific rehabilitation strategies to mitigate strength decline and promote functional independence.

Our results suggest that pinch strength progressively improves over time following surgery, indicating a natural recovery trajectory. This aligns with studies demonstrating gradual neuromuscular adaptation and sensory recovery post-surgery [40,41,42,43]. However, persistent deficits in fine motor control have been reported in long-term survivors, particularly those experiencing chemotherapy-induced peripheral neuropathy (CIPN) [44]. Including time since surgery as a covariate in predictive models for functional recovery could enhance our understanding of rehabilitation timelines. Furthermore, research suggests that early initiation of physical therapy interventions can accelerate functional gains and reduce neuropathic pain [23], reinforcing the importance of structured rehabilitation in the early post-operative phase.

Peripheral neuropathies, often secondary to aggressive cancer treatments (e.g., chemotherapy, radiotherapy, or surgery), can contribute to somatosensory alterations in the upper limb. Such nerve damage can lead to muscle atrophy and strength loss [45,46], which could explain the relationships observed in our results.

Our analysis identified grip strength, pinch strength, and functional sensitivity as predictors of shoulder pain and disability, explaining 25.7% of the variance in perceived disability. These results align with studies indicating that upper limb weakness correlates with increased pain and reduced quality of life in breast cancer survivors [46,47,48]. Reduced muscle strength may contribute to altered movement patterns and compensatory strategies, exacerbating musculoskeletal strain and discomfort. Consequently, interventions targeting muscle strengthening, sensory retraining, and pain management are essential to minimize long-term disability. The integration of task-specific training and progressive resistance exercises has demonstrated efficacy in improving upper limb function post-treatment [49,50].

These findings indicate that strength loss, combined with other sequelae such as reduced mobility, sensitivity, and manual dexterity, generates pain and disability, impacting occupational performance [46,48]. Consequently, recent studies emphasize the need for integrating multiple therapeutic approaches to enhance rehabilitation outcomes in musculoskeletal conditions for patients undergoing breast cancer surgery with axillary dissection. For instance, combining muscle energy techniques with Mulligan mobilization has shown significant improvements in range of motion, posture, and upper-limb functionality for this population.

Although grip strength, manual dexterity, and functional sensitivity explain 25.7% of the risk for developing a disability, numerous studies highlight how these deficits significantly limit the lives of breast cancer survivors, affecting functional tasks, participation, and quality of life [47,48,49]. Deficits in upper-limb functionality have been linked to reduced quality of life, social participation, and the performance of daily activities [36,42].

Therefore, creating specific programs for individuals who have experienced breast cancer could be an optimal solution to address functional, performance, and quality-of-life issues associated with this pathology. Pereira-Rodríguez et al. identified numerous exercise programs specifically designed for this population [49]. Evidence suggests that implementing these programs during the early stages of the disease can prevent upper-limb strength loss and improve mobility [49].

However, the psychological impacts of breast cancer and the associated disorders often go unnoticed, hindering rehabilitation and subsequent recovery [46]. Current scientific evidence demonstrates that the physical, socioemotional, and cognitive consequences of breast cancer significantly affect daily roles and activities. Occupational therapy programs have been shown to enhance specific functional rehabilitation, daily participation, and quality of life for this population [50,51]. Duque et al. [52] emphasized the importance of establishing support networks during the recovery process and the need for future research to solidify the role of occupational therapy in providing comprehensive care for breast cancer survivors.

Employment status was significantly associated with perceived disability, with employed participants reporting lower SPADI scores. This finding suggests that occupational engagement may serve as a protective factor against disability, potentially due to increased physical activity, structured routines, and access to workplace accommodations. Previous studies highlight the psychosocial and physical benefits of returning to work, including improved mental well-being and greater motivation to adhere to rehabilitation [46,47]. However, barriers such as workplace stigma and reduced physical capacity often hinder reintegration [48]. These findings underscore the need for workplace support programs, including flexible work hours and adaptive task modifications, to facilitate successful reintegration and maintain functional independence.

While our results demonstrate significant impairments in upper limb functionality in women with breast cancer and specific predictors explaining their perceived disability, several factors should be considered when applying these findings to other populations.

This study has some methodological limitations. Its cross-sectional design limits the generalizability of the results, which is accompanied by a limited sample size and heterogeneity of the selected sample. Similarly, measurement biases stand out, given that the assessment instruments used were validated in a population different from the study sample. In addition, it should be noted that the current literature on the subject is scarce, so it was difficult to find similar studies or studies in which all the variables we used were evaluated.

These findings highlight the clinical relevance of incorporating comprehensive upper limb assessments into routine care for breast cancer survivors. Early identification of functional deficits can guide personalized rehabilitation strategies aimed at improving strength, sensory function, and dexterity, thereby enhancing quality of life and facilitating a return to daily activities, including work. Future research should focus on longitudinal studies to further explore the long-term impacts of these impairments and to refine intervention programs that address both functional limitations and the psychosocial aspects of recovery.

## 5. Conclusions

In this study, we found that women with breast cancer experience significant impairments in upper limb functionality, particularly in grip strength, digital pinch grip, sensory function, and manual dexterity. Notably, the analysis revealed that age, employment status, and time since surgery influence the degree of disability, with younger women and those still actively working showing different functional profiles from those of older or retired participants. The regression analysis indicated that grip strength, pinch strength, manual dexterity and time since surgery, are key predictors of perceived shoulder pain and disability, explaining 82.3% of the variance in total SPADI scores.

## Figures and Tables

**Table 1 healthcare-13-00766-t001:** Sociodemographic and clinical characteristics of the study participants.

Age	55.9 ± 10.62 (35–89)
Body mass index	25.84 ± 3.94 (16.99–37.17)
Months since surgery	85.90 ± 67.26 (10–300)
Dominant limb was affected	
Yes	21 (50)
No	21 (50)
Dominant hand	
Right	35 (83.3)
Left	6 (14.3)
Ambidextrous	1 (2.4)
Affected limb	
Right	22 (52.4)
Left	20 (47.6)
Work environment	
Active work	21 (50)
Unemployed	10 (23.8)
Retired	9 (21.4)
Disabled	1 (2.4)
Type of work	
Manual	28 (66.7)
Cognitive	12 (28.6)
Underarm surgery	
Yes	34 (81)
No	8 (19)
Chemotherapy	
Yes	30 (71.4)
No	12 (28.6)
Radiotherapy	
Yes	37 (88.1)
No	4 (9.5)
Hormonal therapy	
Yes	28 (66.7)
No	13 (31)

Mean ± standard deviation (minimum–maximum). Frequency (Percentage). Chemotherapy, Radiotherapy and Hormonal therapy are coadjuvant treatments.

**Table 2 healthcare-13-00766-t002:** Descriptive statistics of upper limb functionality tests.

Grip strength in the affected hand	22.50 ± 16.53 (0.50–63.00)
Grip strength in the non-affected hand	24.50 ± 16.26 (1.00–63.00)
Pinch strength in the affected hand	2.3 ± 2.67 (0.10–14.50)
Pinch strength in the non-affected hand	2.86 ± 2.92 (0.10–9.70)
Total NSA Time (stereognosis)	59 ± 22.63 (31.35–153.45)
Total NSA Score (stereognosis)	20.40 ± 1.17 (8.06–20.08)
Moberg O-E (affected hand)	13.07 ± 2.97 (8–24)
Moberg O-E (non-affected hand)	14.18 ± 9.19 (8–69)
Moberg C-E (affected hand)	25.20 ± 6.81 (13–44)
Moberg C-E (non-affected hand)	24.82 ± 7.22 (15–43)
Total JTHFT Time (affected hand)	58.29 ± 17.28 (36–103)
Total JTHFT Score (non-affected hand)	60.43 ± 22.60 (32–149)
Total Pain (SPADI)	17.93 ± 15.32 (0–44)
Total Disability (SPADI)	25.86 ± 22.81 (0–69)
Total SPADI Score	43.79 ± 37.29 (0–111)

Mean ± standard deviation (minimum–maximum). JTHFT: Jebsen–Taylor Hand Function Test; NSA: Nottingham Somatosensory Assessment; O-E: Open Eyes; C-E: Closed Eyes; SPADI: Shoulder Pain and Disability Index.

**Table 3 healthcare-13-00766-t003:** Comparative analysis of upper limb functionality by age groups.

Variables	≤50 Years N = 13	>50 Years N = 29	(F) *p*	Cohen’s *d*
Grip strength in the affected hand	19.98 ± 6.44	23.84 ± 19.47	(11.23) 0.002	−0.232
Pinch strength in the affected hand	2.21 ± 1.70	2.35 ± 3.03	(0.872) 0.356	−0.052
Total NSA Time (stereognosis)	54.02 ± 19.79	62.13 ± 23.13	(0) 0.993	−0.366
Moberg O-E in the affected hand	11.84 ± 2.49	13.58 ± 3.02	(0.097) 0.757	0.603
Moberg C-E in the affected hand	25.53 ± 8.50	25.26 ± 6.56	(0.668) 0.419	0.037
Total JTHFT Time in the affected hand	57.27 ± 14.53	59.23 ± 18.53	(1.029) 0.317	−0.113
Total Pain (SPADI)	20.69 ± 16.52	16.79 ± 14.78	(0.776) 0.384	0.254
Total Disability (SPADI)	28 ± 23.42	25.66 ± 22.39	(0.162) 0.690	0.103
Total Score (SPADI)	48.69 ± 39.11	42.45 ± 36.21	(0.345) 0.560	0.168

Median ± standard deviation. (F): Student’s t; JTHFT: Jebsen–Taylor Hand Function Test; NSA: Nottingham Somatosensory Assessment; O-E: Open Eyes; C-E: Closed Eyes; *p*: *p*-value; SPADI: Shoulder Pain and Disability Index.

**Table 4 healthcare-13-00766-t004:** Comparative analysis of upper limb functionality by employment status.

Variables	Working N = 21	Non-Working N = 21	(F) *p*	Cohen’s *d*
Grip strength in affected hand	22.9 ± 16.03	22.40 ± 17.47	(0.100) 0.754	0.030
Pinch strength in affected hand	2.38 ± 1.80	2.23 ± 3.37	(0.879) 0.354	0.055
Total NSA Time (Stereognosis)	59.30 ± 16.84	49.94 ± 27.02	(0.627) 0.433	−0.029
Moberg O-E in affected hand	12.99 ± 3.43	13.09 ± 2.45	(0.795) 0.378	−0.035
Moberg C-E in affected hand	24.96 ± 7.64	25.73 ± 6.75	(0.156) 0.695	−0.106
Total JTHFT Time in affected hand	55.33 ± 15.11	61.94 ± 18.92	(0.393) 0.535	−0.385
Total Pain (SPADI)	14.57 ± 13.43	21.43 ± 16.48	(2.586) 0.116	−0.456
Total Disability (SPADI)	20.52 ± 17.66	32.24 ± 25.48	(6.472) 0.015	−0.534
Total Score SPADI	35.10 ± 30.16	53.67 ± 40.99	(5.316) 0.026	−0.516

Mean ± standard deviation. (F): Student’s t; JTHFT: Jebsen–Taylor Hand Function Test; NSA: Nottingham Somatosensory Assessment; O-E: Open Eyes; C-E: Closed Eyes; *p*: *p*-value; SPADI: Shoulder Pain and Disability Index.

**Table 5 healthcare-13-00766-t005:** Comparative analysis of upper limb functionality by time since surgery.

Variables	≤4 Years N = 16	>4 Years N = 17	(F) *p*	Cohen’s *d*
Grip strength in affected hand	16.88 ± 8.42	14.06 ± 6.66	(1.343) 0.225	0.373
Pinch strength in affected hand	1.60 ± 1.33	1.07 ± 0.91	(5.419) 0.027	0.467
Total NSA Time (stereognosis)	55.68 ± 18	53.26 ± 13.86	(0.253) 0.618	0.151
Moberg O-E in affected hand	13.32 ± 3.63	13.33 ± 2.83	(0.001) 0.976	−0.004
Moberg C-E in affected hand	26.04 ± 8.73	24.02 ± 5.72	(4.013) 0.054	−0.275
Total JTHFT Time	53.50 ± 14.88	63.39 ± 19.66	(2.024) 0.165	−0.565
Total Pain (SPADI)	17.13 ± 15.54	23 ± 14.87	(0.001) 0.973	−0.387
Total Disability (SPADI)	29.06 ± 23.91	30.94 ± 21.44	(0.205) 0.654	−0.083
Total SPADI Score	46.19 ± 38.76	53.94 ± 35.19	(0.345) 0.561	−0.210

Mean ± standard deviation. (F): Student’s t; JTHFT: Jebsen–Taylor Hand Function Test; NSA: Nottingham Somatosensory Assessment; O-E: Open Eyes; C-E: Closed Eyes; *p*: *p*-value; SPADI: Shoulder Pain and Disability Index.

**Table 6 healthcare-13-00766-t006:** Regression model summary for predictor variables of disability.

Model	R	R^2^	R^2^ Adjusted	Standard Error of Estimation
1	0.922 ^a^	0.823	0.083	0.933

^a^ Predictor Variables: (Constant), Pinch Strength, Moberg Pick-Up Test in affected hand with eyes closed, total time Jebsen–Taylor Hand Function Test in affected hand, and Grip Strength, and Time Since Surgery.

**Table 7 healthcare-13-00766-t007:** ANOVA results for the regression model.

Model		Sum of Squares	gl	Quadratic Mean	F	Sig.
1	Regression ^a^	40.915	5	8.183	9.393	0.025 ^b^
	Waste	3.485	4	0.871		
	Total	44.400	9			

^a^ Predictor Variables: (Constant), Pinch Strength, Moberg Pick-Up Test in affected hand with eyes closed, total time Jebsen–Taylor Hand Function Test in affected hand, and Grip Strength and Time Since Surgery. ^b^ Dependent variable: Total SPADI.

## Data Availability

Data are available by contacting the authors.

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
