# Peer review of "Impact of Unilateral Breast Cancer Surgery on Upper Limb Functionality: Strength, Manual Dexterity, and Disability Prediction"

_healthcare, 2025, doi:10.3390/healthcare13070766_

Round 1

Reviewer 1 Report

Comments and Suggestions for Authors

- Numerous grammatical errors hinder readability. It is recommended that a native English editor revise the manuscript to enhance clarity.

Title

- The title should be revised to more accurately reflect the study's aims.

Abstract

- The abstract requires a complete revision. It should briefly explain the necessity for conducting the study in the “objectives” section. Mentioning the centers from which participants were recruited is unnecessary; instead, focus on important aspects of the methods.

Introduction

- Although the authors assert that no occupational therapy studies have examined upper limb functionality in breast cancer patients, numerous relevant studies exist in the literature that address this topic, regardless of whether they were conducted by occupational therapists or other professionals. For instance, the study titled 'Upper extremity strength and range of motion and their relationship to function in breast cancer survivors' (Physiotherapy Theory and Practice, 2013) provides valuable insights into this area. It is essential for the authors to acknowledge these existing studies in their introduction, as this will enhance the context and significance of their research.

- The introduction should clearly articulate the necessity of the current study and specify how it contributes to existing knowledge in the field. Additionally, the aims of the study should be precisely stated in the final paragraph of the introduction.

Methodology

- The rationale for dividing samples into groups based on age, years since surgery, and employment status is unclear.

- While the results are provided separately for the dominant and non-dominant hands in Table 2, other tables present the results for affected and non-affected hands. This inconsistency may confuse readers.

- Replace "Dynamometry" and "Pinch dynamometry" with “Grip strength” and “Pinch strength” throughout the manuscript.

- Clarification is needed regarding what functional aspects of upper extremity are evaluated by the Moberg Pick-up Test and its validity in breast cancer patients.

- Page 3, lines 107-111 contains misleading information about the “Shoulder Pain and Disability Index,” which has two subscales; clarity is needed on this point.

- Additionally, what does “MS” refer to on page 3, line 109?

-  Provide reliability and validity data for all measures used in this study concerning breast cancer patients.

- Was there a rest interval between assessments?

-  The statistical analysis should use stepwise regression models (R²). Initially assess correlations using Pearson or Spearman coefficients to determine relevant factors for inclusion (see Iranian Rehabilitation Journal, 2018).

- Were data from different assessments normally distributed? What statistical tests were used?

Results

- Page 5, lines 144-148: Page 5, lines 144-148 is redundant since descriptive data are already presented in Table 2; it should be replaced with a brief reference to Table 2.

- The title of Table 2 should clearly describe its contents. This comment applies to Tables 3-7 as well.

- Page 7, lines 161-162: How was the activity level of the participants measured, and how were the participants classified into active and passive groups?

- The discussion should be exclusively based on the findings of the current study, rather than on unrelated topics. Therefore, it requires a complete revision and rewriting. For instance, the comparison between dominant and non-dominant hands in different parameters was not an aim of this study and was not conducted. Additionally, activities of daily living (ADLs) were not directly assessed in this research. Consequently, the second paragraph of the discussion is not relevant to the results. After addressing the statistical analysis based on my previous comments, the discussion should focus on the correlation between the disability index and various upper extremity measures, including grip strength, pinch strength, and manual dexterity, as well as the clinical relevance of the study's findings.

- Does “gripper strength” refer to “pinch strength”? If so, please replace it throughout the text. Additionally, clarification is needed regarding what is meant by "pincer strength."

Conclusion

- The conclusion should be rewritten to first summarize findings before discussing their clinical relevance.

Overall, addressing these points will significantly improve the manuscript's quality and clarity. Thank you for considering my comments.

Comments on the Quality of English Language

Numerous grammatical errors hinder readability. It is recommended that a native English editor revise the manuscript to enhance clarity.

Author Response

Dear Reviewer
We would like to express our sincere gratitude for your thorough review and insightful comments. All the suggested feedback has been carefully considered and implemented, leading to significant improvements in the cohesion and scientific rigor of the study.
These enhancements have strengthened the manuscript, positioning it as a more valuable contribution to both the academic and clinical communities.
Please find below our detailed responses to each of your comments.
Kind regards

Comment 1:  Numerous grammatical errors hinder readability. It is recommended that a native English editor revise the manuscript to enhance clarity
Reply to the comment 1.
The article's editing has been reviewed by a professional company specializing in scientific and healthcare translation. The document has been resubmitted, and some errors identified have been corrected. A verification certificate from ServingMED, a professional scientific and healthcare translation company, is included

Comment 2: Title: The title should be revised to more accurately reflect the study's aims

Reply to the comment 2.
Thank you for pointing this out. I/We agree with this comment. We have changed the title of the study: "Impact of Unilateral Breast Cancer Surgery on Upper Limb Functionality: Strength, Manual Dexterity, and Disability Prediction”. 
Updated text in the manuscript in Page 1, Paragraph 1 and line 2-4

Comment 3: Abstract: The abstract requires a complete revision. It should briefly explain the necessity for conducting the study in the “objectives” section. Mentioning the centers from which participants were recruited is unnecessary; instead, focus on important aspects of the methods.
Reply to the comment 3. 
Thank you for pointing this out. I/We agree with this comment. Therefore, We have completely revised the abstract based on your recommendations. Updated text in the manuscript in Page 1, lines 14-31.
Comment 4: Introduction:  Although the authors assert that no occupational therapy studies have examined upper limb functionality in breast cancer patients, numerous relevant studies exist in the literature that address this topic, regardless of whether they were conducted by occupational therapists or other professionals. For instance, the study titled 'Upper extremity strength and range of motion and their relationship to function in breast cancer survivors' (Physiotherapy Theory and Practice, 2013) provides valuable insights into this area. It is essential for the authors to acknowledge these existing studies in their introduction, as this will enhance the context and significance of their research
Reply to the comment 4: 
We agree and are grateful for this comment. Therefore, I/We have incorporated the suggested reference into the introduction
Updated text in the manuscript in Page 2 Paragraph 5 and line 63-69
Comment 5:  The introduction should clearly articulate the necessity of the current study and specify how it contributes to existing knowledge in the field. Additionally, the aims of the study should be precisely stated in the final paragraph of the introduction.
Reply to the comment 5: We appreciate the reviewer’s insightful feedback. In response, we have revised the introduction to better articulate the necessity of this study and its contribution to the field. Additionally, we have reformulated the aims of the study in the final paragraph of the introduction
Updated text in the manuscript in Page 2, Paragraph 6-7 and line 74-90
Methodology

Comment 6: - The rationale for dividing samples into groups based on age, years since surgery, and employment status is unclear
Reply to the comment. 6: 
Thank you for pointing this out. I/We agree with this comment. Therefore, I/we have added the necessary information to clarify this aspect.
Updated text in the manuscript in Page 3 Paragraph 2-6 and line 107-133
Comment 7: While the results are provided separately for the dominant and non-dominant hands in Table 2, other tables present the results for affected and non-affected hands. This inconsistency may confuse readers.
Reply to the comment 7.  
We acknowledge the reviewer’s comments and have made the requested modifications. Additionally, the tables and corresponding data have been updated to ensure consistency throughout the article. These changes have been made to enhance the coherence and scientific rigor of the study. 
Updated text in the manuscript in Pages 6-7 (Table 2). 
Comment 8: Replace "Dynamometry" and "Pinch dynamometry" with “Grip strength” and “Pinch strength” throughout the manuscript
Reply to the comment 8: 
Thank you for pointing this out. We replace  text in the manuscript  "Dynamometry" and "Pinch dynamometry" with “Grip strength” and “Pinch strength” throughout the manuscript.
Comment 9: Clarification is needed regarding what functional aspects of upper extremity are evaluated by the Moberg Pick-up Test and its validity in breast cancer patients
Reply to the comment.  9: We acknowledge the reviewer’s comments and have made the requested modifications.
Updated text in the manuscript in Page 4, point 3 (Moberg Pick-up Test), line 159-175
Comment 10: Page 3, lines 107-111 contains misleading information about the “Shoulder Pain and Disability Index,” which has two subscales; clarity is needed on this point.
Reply to the comment 10: I/We agree and are grateful for this comment. Updated text in the manuscript in Page 4, point 5 and line 190-193.
Comment 11: Additionally, what does “MS” refer to on page 3, line 109?
Reply to the comment.  11: It is upper limb. Updated in the manuscript in Page 4, point 5 and line 188.
Comment 12: Provide reliability and validity data for all measures used in this study concerning breast cancer patients
Reply to the comment.  12: We acknowledge the reviewer’s comments and have made the requested modifications in all the outcome measures, which are presented below and can be verified in the Data Collection section 2.2. (line 143-199).
•    Grip Strength Measurement: Page 4, paragraph 1, line 143-47
•    Nottingham Sensory Assessment (Stereognosis Subscale). Page 4, point 2, line 152-153
•    Moberg Pick-up Test (Manual Dexterity and Sensory Function). Page 4, point 3, line 175-78.
•    Jebsen-Taylor Hand Function Test (JTHFT)- Page 4, point 4, line 183-185.
•    Shoulder Pain and Disability Index (SPADI): Page 4, point 5, line 190-193
•    Pinch strength. Pag 5, point 6, line 200 and page 5, line 197-199.

Comment 13: Was there a rest interval between assessments?
Reply to the comment.  13: I/We agree and are grateful for this comment .All assessments were conducted in a single session, with a short rest period between each. Updated text in the manuscript in Page 3, line 136. 
Comment 14: The statistical analysis should use stepwise regression models (R²). Initially assess correlations using Pearson or Spearman coefficients to determine relevant factors for inclusion (see Iranian Rehabilitation Journal, 2018)
Reply to the comment.  14: I/We agree and are grateful for this comment. That type of analysis was carried out. We have modified the text so that it is better understood. Not all tables are included due to the word limitation of the article, but this statistical process was developed.
Comment 15: Were data from different assessments normally distributed? What statistical tests were used?
Reply to the comment 15: I/We agree and are grateful for this comment: The distribution was normal for the sample studied in the variables analyzed. Updated text in the manuscript in Page 5, into Data Analysis, line 204-205.
Results
Comment 16: Page 5, lines 144-148 is redundant since descriptive data are already presented in Table 2; it should be replaced with a brief reference to Table 2
 Reply to the comment 16: I/We agree and are grateful for this comment: The text is changed, with the proposed modification.
Comment 17: The title of Table 2 should clearly describe its contents. This comment applies to Tables 3-7 as well.
Reply to the comment 17: I/We agree and are grateful for this comment: Changed the text of the titles of all tables to make them more descriptive. Updated text in the manuscript in Page 6-7, table 2.
Comment 18: Page 7, lines 161-162: How was the activity level of the participants measured, and how were the participants classified into active and passive groups?
Reply to the comment 18. I/We agree and are grateful for this comment:
In this study, the "active" group refers to participants who are currently employed and engaged in work activities. In contrast, the "passive" group includes individuals who are not working, either due to unemployment, retirement, recognized disability, or an existing occupational incapacity (i.e., being on medical leave). This classification allows for the assessment of how employment status influences upper limb functionality and perceived disability.
Comment 19: The discussion should be exclusively based on the findings of the current study, rather than on unrelated topics. Therefore, it requires a complete revision and rewriting. For instance, the comparison between dominant and non-dominant hands in different parameters was not an aim of this study and was not conducted. Additionally, activities of daily living (ADLs) were not directly assessed in this research. Consequently, the second paragraph of the discussion is not relevant to the results. After addressing the statistical analysis based on my previous comments, the discussion should focus on the correlation between the disability index and various upper extremity measures, including grip strength, pinch strength, and manual dexterity, as well as the clinical relevance of the study's findings
Reply  to the comment 19: I/We agree and are grateful for this comment. We have constructed a new discussion based on your generous comments and the new references utilized.
Updated text in the manuscript in Page 10-12. 
Comment 20: -Does “gripper strength” refer to “pinch strength”? If so, please replace it throughout the text. Additionally, clarification is needed regarding what is meant by "pincer strength.
Reply to the comment 20: I/We agree and are grateful for this comment.
The “gripper strength” refer to “pinch strength” we change in the text
Conclusion:
Comment 21:  The conclusion should be rewritten to first summarize findings before discussing their clinical relevance
Reply to the comment. 21: I/We agree and are grateful for this comment. We have rewritten this section following your valuable comments.
Updated text in the manuscript in Page 12 (5. Conclusion) 

Reviewer 2 Report

Comments and Suggestions for Authors

 The current manuscript aims to study the impact of breast cancer on upper limb functionality such as grip strength, digital pinch grip, sensitivity, and manual dexterity. The manuscript in the current form cannot be publishable. Please find the following comments:

A- Introduction:
1- Redefine the cancer with emphasis on breast cancer

2- Line 59-60: you mention that ‘ no occupational therapy research has yet analyzed upper limb functionality.’’ . The conducted study is cross-sectional study, not interventional study. This statement is out of the context.

B- Methods:
1- As the inclusion criteria including women who had undergone surgery for unilateral breast cancer. This should be clear in the title and introduction.

2- The inclusion criteria should be clearer in terms of type of surgery, and upper limb neuropathy.

3- The first item in the exclusion criteria equals the first item in the inclusion criteria.

4- Line 109: What does the NS abbreviation refer to?

5- Line 112: What does the JAMAR abbreviation refer to?

6- Report your study according to the STROBE reporting guidelines

7- Discuss the sample size calculation

C- Results:
1- How did you include a participant without surgery. It is mentioned in the inclusion criteria
2- compare between the affected and non-affected limb instead of dominant and non-dominant side
3- The regression model did not show any significant variable or predictor

D- Discussion:
1- You can recommend based on your results in the importance of rehabilitation interventions. There is a good example (https://doi.org/10.3390/jcm13040980 )

Comments on the Quality of English Language

The English could be improved to more clearly express the research.

Author Response

Dear Reviewer
We would like to express our sincere gratitude for your thorough review and insightful comments. All the suggested feedback has been carefully considered and implemented, leading to significant improvements in the cohesion and scientific rigor of the study.
These enhancements have strengthened the manuscript, positioning it as a more valuable contribution to both the academic and clinical communities.
Please find below our detailed responses to each of your comments.
Kind regards

Introduction:
Comments 1: Redefine the cancer with emphasis on breast cancer
Reply to the comment 1.  Thank you for pointing this out. I/We agree with this comment. Therefore, We have removed and replaced it with the following information.
Updated text in the manuscript in Page 2, Paragraph 2-3 and line 44-54.

Comments 2: Line 59-60: you mention that ‘ no occupational therapy research has yet analyzed upper limb functionality.’’ . The conducted study is cross-sectional study, not interventional study. This statement is out of the context.
Reply to the comment 2:  Thank you for pointing this out. I/We agree with this comment.  The text is changed, with the proposed modification.

Methods:
Comments 1: As the inclusion criteria including women who had undergone surgery for unilateral breast cancer. This should be clear in the title and introduction
Reply to the comment 1: Thank you for pointing this out. I/We agree with this comment.  
We have changed the title based on your recommendation. Updated text in the manuscript in Page 1, Paragraph 1 and line 2-4. We modified the introduction, also incorporating suggestions from other reviewers.
Comments 2:  The inclusion criteria should be clearer in terms of type of surgery, and upper limb neuropathy
Reply to the comment 2. The text is changed, with the proposed modification.
Updated in the manuscript in Page 3, line 97-98.
Comments 3: The first item in the exclusion criteria equals the first item in the inclusion criteria.
Reply to the comment.  3: Both are different because the inclusion criterion focuses on women with unilateral breast cancer who underwent surgery, while the exclusion criterion excludes those with bilateral breast cancer, which is a different condition.
Comments 4- Line 109: What does the NS abbreviation refer to?
Reply to the comment 4: It is upper limb. Updated in the manuscript in Page 4, point 5 and line 188.
Comments 5- Line 112: What does the JAMAR abbreviation refer to?
Reply to the comment 5: Thank you for your comment. JAMAR refers to the registered trademark of the instrument's name. It will be modified to Jamar®
Updated in the manuscript in Page 3, last paragraph, line 139 and page 4, last paragraph, line 194.
 Comments 6- Report your study according to the STROBE reporting guidelines
Reply to the comment 6: We agree and are grateful for this comment. The text is changed, with the proposed modification
Comments 7- Discuss the sample size calculation
Reply to the comments 7: The sample size could not be calculated because there is not previous studio to be done. This study would be a Good starting point to make a serius sample size calculation

C-Results:
Comments 1- How did you include a participant without surgery. It is mentioned in the inclusion criteria
Reply to the comment  1:  We are sorry, it was a mistake We change the text. All participants underwent surgery. We removed this row from the table
Comments 2- compare between the affected and non-affected limb instead of dominant and non-dominant side
Reply to the comment 2. Thank you for pointing this out. We have made this change in the text and in the table 2
Comments 3- The regression model did not show any significant variable or predictor. 
Reply to the comment 3: Thank you for pointing this out. The regression model has been modified in response to the editor's comments. See tables 6-7 (page 10).

D- Discussion:
Comments 1- You can recommend based on your results in the importance of rehabilitation interventions. There is a good example (https://doi.org/10.3390/jcm13040980 )
Reply to the comment: Thank you for pointing this out. I/We agree with this comment  
The discussion has been completely revised; however, the change addressing your suggestion can be found in the manuscript in Page 11,  paragraph 1-2 and line 287-298.

Comments on the Quality of English Language. 
The article's editing has been reviewed by a professional company specializing in scientific and healthcare translation. The document has been resubmitted, and some errors identified have been corrected. A verification certificate from ServingMED, a professional scientific and healthcare translation company, is included.

Reviewer 3 Report

Comments and Suggestions for Authors

First of all, congratulations to the research team for the area of ​​post-mastectomy women. The aspects listed are intended to add value to the article.

-Introduction: the topic of breast cancer is addressed, but some reference must be made to the types of treatments to understand the context of the study.

-Methodology: the inclusion criteria are unclear whether chemotherapy, radiotherapy or hormone therapy are co-adjuvant therapies to surgery.

-Discussion: the importance of rehabilitation programs is mentioned, but it is not described in the introduction or in the data collected, so it must be analyzed.

-The references presented are old, considering the evolution of scientific production in the area.

Author Response

Dear Reviewer
We would like to express our sincere gratitude for your thorough review and insightful comments. All the suggested feedback has been carefully considered and implemented, leading to significant improvements in the cohesion and scientific rigor of the study.
These enhancements have strengthened the manuscript, positioning it as a more valuable contribution to both the academic and clinical communities.
Please find below our detailed responses to each of your comments.
Kind regards
Comments 1: Introduction: 
The topic of breast cancer is addressed, but some reference must be made to the types of treatments to understand the context of the study.
Reply to the comment: Thank you for pointing this out. I/We agree with this reply. We have incorporated the following updated information regarding the treatment provided to these patients and its relationship with occupational therapy.
Updated in the manuscript in Page 2, paragraph 3, line 49-55 and paragraph 6-7, line 74-90.
Comments 2: Methodology: the inclusion criteria are unclear whether chemotherapy, radiotherapy or hormone therapy are co-adjuvant therapies to surgery. 
Reply to the comment: Thank you for pointing this out. The text is changed, with the proposed modification in page 6, table 1 (table note)
Comments 3: Discussion: the importance of rehabilitation programs is mentioned, but it is not described in the introduction or in the data collected, so it must be analyzed
Reply to the comment. We agree with this comment. Therefore, we have completely revised the introduction and discussion based on your recommendations and those of the other reviewers. Please review the introduction and discussion to ensure that your suggestions have been taken into account.
Comment 4. The references presented are old, considering the evolution of scientific production in the area 
Reply to the comment. Thank you for pointing this out. I/We agree with this comment. We have included several recent references, which can be verified in the bibliography

Reviewer 4 Report

Comments and Suggestions for Authors

The article examines differences in upper limb functionality among women with breast cancer, focusing on grip strength, digital pinch, sensitivity, and manual dexterity, as well as their relationship to perceived disability. The study employs a cross-sectional and descriptive design, evaluating 42 participants using tools such as dynamometry and the SPADI index. The results indicate that factors such as age, time since surgery, and employment status significantly influence upper limb functionality. Specifically, older women, those not actively employed, or those further from their surgical intervention show reduced strength and manual dexterity. Furthermore, grip strength, digital pinch, manual function, and sensitivity account for 25.7% of the variability in perceived pain and disability. The study concludes that breast cancer causes significant motor and sensory impairments in the affected upper limb, impacting the quality of life of the affected individuals.

However, the article presents significant limitations that must be addressed to enhance its scientific and methodological quality. Firstly, the introduction is overly brief and lacks cohesion, failing to provide a robust theoretical framework. A more comprehensive review of effective therapies for upper limb rehabilitation should be included, emphasizing the role of Occupational Therapy to better contextualize the study’s objectives and clinical relevance.

In the “Study Design and Sampling” section, greater clarity regarding the composition of the study groups is needed. Details such as the number of groups formed, the average age of the participants, and a more thorough characterization of the sample—including socioeconomic status and physical activity level—are essential to better understand the observed differences between groups.

Another critical issue concerns the evaluation of intra- and inter-observer reliability. While the article mentions that dynamometry has good reliability, it does not detail how this was assessed. It is crucial to describe the number of observers involved, their level of expertise in administering the tests, and the methodology used to evaluate reliability. This information should also extend to other tools employed in the study, such as the SPADI index and functional tests. Without these details, the validity of the results is questionable.

Lastly, the data analysis section states that the t-test was used, but it does not mention whether the normality of the sample was tested beforehand. To ensure the transparency and robustness of the statistical analyses, it is necessary to include the results of normality tests, such as the Shapiro-Wilk test, and explain how these tests were conducted in the methodology section.

In conclusion, while the article addresses a relevant topic and provides interesting findings, significant improvements in its structure, theoretical justification, and methodology are required. Implementing these changes will enhance the scientific rigor of the study and position it as a stronger and more valuable contribution to the academic and clinical community.

Author Response

Dear Reviewer
We would like to express our sincere gratitude for your thorough review and insightful comments. All the suggested feedback has been carefully considered and implemented, leading to significant improvements in the cohesion and scientific rigor of the study.
These enhancements have strengthened the manuscript, positioning it as a more valuable contribution to both the academic and clinical communities.
Please find below our detailed responses to each of your comments.
Kind regards
Comments 1: Firstly, the introduction is overly brief and lacks cohesion, failing to provide a robust theoretical framework. A more comprehensive review of effective therapies for upper limb rehabilitation should be included, emphasizing the role of Occupational Therapy to better contextualize the study’s objectives and clinical relevance. 
Reply to the comment 1: Thank you for pointing this out. We agree with this comment. Therefore, we have expanded and updated the information in the introduction. In red, we have highlighted changes based on your comments, but we have also included other changes suggested by other reviewers. The new wording provides a more cohesive and robust perspective.
Updated in the manuscript in Page 2, paragraph 1-3, line 38-55; paragraph 5, line 63-69 and paragraph 6-7, line 74-90.
Comments 2: In the “Study Design and Sampling” section, greater clarity regarding the composition of the study groups is needed. Details such as the number of groups formed, the average age of the participants, and a more thorough characterization of the sample—including socioeconomic status and physical activity level—are essential to better understand the observed differences between groups.
Reply to the comment 2. Thank you for pointing this out. We agree with this comment. The text is changed, with the proposed modification, you can review the change in page 3 (2.1. Study Desing an  Sampling), paragraph 2-6, lines 110-133.
Comments 3: Another critical issue concerns the evaluation of intra- and inter-observer reliability. While the article mentions that dynamometry has good reliability, it does not detail how this was assessed. It is crucial to describe the number of observers involved, their level of expertise in administering the tests, and the methodology used to evaluate reliability. This information should also extend to other tools employed in the study, such as the SPADI index and functional tests. Without these details, the validity of the results is questionable. 
Reply to the comment 3: Thank you for pointing this out. We agree with this comment. We explain more all the aspects of Data Collection. 
Updated in the manuscript in Page 3, line 137; Page 3, point 1 (grip strength), line 139-143); Page 4, point 2, line 149-152; Page 4, point 3, line 155-158; Page 4, point 4, line 179-182; Page 4, point 5, line 186-189 and Page 4-5, point 6, line 194-197.
Comments 4: Lastly, the data analysis section states that the t-test was used, but it does not mention whether the normality of the sample was tested beforehand. To ensure the transparency and robustness of the statistical analyses, it is necessary to include the results of normality tests, such as the Shapiro-Wilk test, and explain how these tests were conducted in the methodology section
Reply to the comment 4: Thank you for pointing this out. We agree with this comment.
Updated in the manuscript in Page 5, In the first paragraph of Data Analysis, line 204-205.

Round 2

Reviewer 1 Report

Comments and Suggestions for Authors

Dear Authors,

Thank you for submitting your revised manuscript. While improvements are evident, the Introduction and Discussion sections require further restructuring to align with the journal’s standards. Below are my detailed suggestions:
Introduction
- The introduction provides a general overview of breast cancer and its complications. However, its impact would be strengthened by immediately focusing on the functional consequences of breast cancer surgery, such as upper limb dysfunction. Starting with this issue would engage readers and clarify your study’s purpose.
- It is important to contextualize your research within the existing literature on upper limb functionality in breast cancer survivors. For example, studies such as McNeely et al. (2013) have examined strength and range of motion in this population. To strengthen your rationale, please clarify how your work builds upon or differs from these previous findings. What specific gaps does your study address? For example, are you addressing a lack of data on manual dexterity or the absence of predictive models for disability in this context?
- The transition from general cancer biology to treatment side effects is somewhat abrupt. To improve the flow, consider structuring the narrative as follows: 
1.    Establish the clinical significance of upper limb dysfunction following surgery.
2.    Summarize existing knowledge on this topic, referencing relevant studies beyond those focused solely on occupational therapy.
3.    Identify unresolved questions, such as the specific predictive factors for disability within your study population.
- The concluding paragraph would be stronger if it directly connects your study’s goals to the gaps you’ve identified. For example, something like:
“As breast cancer survival rates increase, effective management of upper extremity dysfunction becomes an increasingly vital component of comprehensive cancer care. This research project addresses this need by analyzing differences in affected upper extremity functionality (grip and pinch strength, functional sensitivity, and manual dexterity) in a Spanish population of breast cancer survivors. Furthermore, it seeks to determine the predictive capacity of these factors on the disability perceived by these women, thus informing targeted interventions to improve their quality of life.”

Materials and Methods
•    Page 3, Lines 107–133: The rationale for grouping participants by age, employment, and time since surgery is clear. However, introduce these variables in the Introduction to emphasize their relevance.
•    Page 4, Line 144: Replace reference #28 with a citation directly supporting the Moberg Pick-Up Test in breast cancer populations.
•    Page 4, Lines 144–146: Revise the sentence “Reliability: High intra- and inter-rater…” for clarity. For example:

“The Moberg Pick-Up Test demonstrates high intra- and inter-rater reliability for assessing functional hand impairments in breast cancer survivors.”
•    Page 4, Lines 144–146: The statement “The validity of the Moberg Pick-up Test in populations with breast cancer has been supported by its reliability and simplicity in assessing functional impairments linked to neuropathic symptoms” is inaccurate. If the reliability and validity of the Moberg test have been directly evaluated in subjects with breast cancer, this should be briefly described, with appropriate references. Additionally, the current explanation of the Moberg test is somewhat redundant and includes repeated information. 
•    Provide references for the reliability and validity of all measures (e.g., pinch/grip strength) specifically in breast cancer populations.
•    Page 5, Line 199: Include a reference validating the pinch strength test.
•    Page 4, Lines 186–188: The sentence “A questionnaire on pain …” is grammatically unclear and should be rephrased.

Results
While you were asked to provide results for the affected versus non-affected hand, Table 2 presents results for dominant and non-dominant hands (i.e. the results of Moberg and JTHFT). Please clarify this discrepancy.

Discussion
I regret to inform you that the discussion section still does not adequately address my previous concerns. The primary issue remains a lack of focus on your own findings and overreliance on tangential topics (e.g., general rehabilitation strategies, psychological impacts, or unrelated studies). Your study has four key findings: 
•    The effect of age on grip strength.
•    The relationship between time since surgery and pinch strength recovery.
•    The association between employment status and disability (i.e., SPADI score).
•    The predictors of shoulder pain and disability.
I suggest the first paragraph of the discussion should briefly reiterate the study's purpose and summarize these key findings. Subsequent paragraphs should then be dedicated to each key finding, thoroughly exploring their implications and comparing them with relevant studies. Extraneous topics should be avoided. 
Furthermore, a separate paragraph should address the study’s limitations (e.g., cross-sectional design, small sample size, measurement tools, and unmeasured confounders such as rehabilitation protocols, psychosocial status, or comorbidities). 
Finally, the discussion should conclude with the clinical implications of your study and suggestions for future research, grounded in your findings. For example, emphasize the importance of post-surgical monitoring of upper limb functionality, particularly grip strength and fine motor skills. 
I urge you to carefully revisit the Discussion section to ensure it directly interprets your results and avoids unsupported extrapolations. As it stands, the current version risks rejection due to its lack of focus. 

Conclusion
- The final paragraph of the conclusion overlaps with the Discussion. Integrate its content into the Discussion’s clinical implications or limitations sections.
- Line 342: “SPADI scores” should be “total SPADI score”.

Incorporating these revisions will significantly enhance readers' understanding of the study's necessity, novelty, and relevance. 

Thank you again for your continued efforts.

Best regards,

Comments on the Quality of English Language

The manuscript would benefit from thorough proofreading to improve grammar, sentence structure, and overall clarity.

Author Response

Dear Reviewer,
We would like to express our sincere gratitude for your thorough review and insightful comments. All the suggested feedback has been carefully considered and implemented, leading to significant improvements in the cohesion and scientific rigor of the study.
These enhancements have strengthened the manuscript, positioning it as a more valuable contribution to both the academic and clinical communities.
Please find below our detailed responses to each of your comments.
Kind regards

Comments 1: Introduction - The introduction provides a general overview of breast cancer and its complications. However, its impact would be strengthened by immediately focusing on the functional consequences of breast cancer surgery, such as upper limb dysfunction. Starting with this issue would engage readers and clarify your study’s purpose. 
It is important to contextualize your research within the existing literature on upper limb functionality in breast cancer survivors. For example, studies such as McNeely et al. (2013) have examined strength and range of motion in this population. To strengthen your rationale, please clarify how your work builds upon or differs from these previous findings. What specific gaps does your study address? For example, are you addressing a lack of data on manual dexterity or the absence of predictive models for disability in this context? 
Response 1: Thank you for pointing this out. We agree with this comment. We have changed
All newly added or modified text has been marked in green for clarity
Updated text in the manuscript in Page 2 Paragraph 5 and line 58-64

Comments 2: The transition from general cancer biology to treatment side effects is somewhat abrupt. To improve the flow, consider structuring the narrative as follows: 1. Establish the clinical significance of upper limb dysfunction following surgery. 2. Summarize existing knowledge on this topic, referencing relevant studies beyond those focused solely on occupational therapy. 3. Identify unresolved questions, such as the specific predictive factors for disability within your study population. 
Response 2: Thank you for pointing this out. We agree with this comment. We have changed. Updated text in the manuscript in Page 2 Paragraph 6 and line 68-77

Comments 3: The concluding paragraph would be stronger if it directly connects your study’s goals to the gaps you’ve identified. For example, something like: “As breast cancer survival rates increase, effective management of upper extremity dysfunction becomes an increasingly vital component of comprehensive cancer care. This research project addresses this need by analyzing differences in affected upper extremity functionality (grip and pinch strength, functional sensitivity, and manual dexterity) in a Spanish population of breast cancer survivors. Furthermore, it seeks to determine the predictive capacity of these factors on the disability perceived by these women, thus informing targeted interventions to improve their quality of life.” 
Response 3: Thank you for pointing this out. We agree with this comment. We have changed
Updated text in the manuscript in Page 2 Paragraph 7 and line 87-93

Comments 4: Materials and Methods : • Page 3, Lines 107–133: The rationale for grouping participants by age, employment, and time since surgery is clear. However, introduce these variables in the Introduction to emphasize their relevance. 
Response 4: Thank you for pointing this out. We agree with this comment. We have changed
Updated text in the manuscript in Page 2 Paragraph 6 and line 68-77

Comments 5: Page 4, Line 144: Replace reference #28 with a citation directly supporting the Moberg Pick-Up Test in breast cancer populations. 
Response 5: Thank you for pointing this out. We agree with this comment. We have changed
Updated text in the manuscript in Page 4 Paragraph 3 and line 162-180

Comments 6:  Page 4, Lines 144–146: Revise the sentence “Reliability: High intra- and inter-rater…” for clarity. For example: “The Moberg Pick-Up Test demonstrates high intra- and inter-rater reliability for assessing functional hand impairments in breast cancer survivors.” 
Response 6:  Thank you for pointing this out. We agree with this comment. We have changed
Updated text in the manuscript in Page 4 Paragraph 5 and line 177-179

Comments 7: • Page 4, Lines 144–146: The statement “The validity of the Moberg Pick-up Test in populations with breast cancer has been supported by its reliability and simplicity in assessing functional impairments linked to neuropathic symptoms” is inaccurate. If the reliability and validity of the Moberg test have been directly evaluated in subjects with breast cancer, this should be briefly described, with appropriate references. Additionally, the current explanation of the Moberg test is somewhat redundant and includes repeated information. • Provide references for the reliability and validity of all measures (e.g., pinch/grip strength) specifically in breast cancer populations. 
Response 7: Thank you for pointing this out. We agree with this comment. We have changed all the text for the Moberg Pick up Test
Updated text in the manuscript in Page 4 Paragraph 5 and line 158-179

Comments 8: • Page 5, Line 199: Include a reference validating the pinch strength test. 
Response 8: The reference for the pinch strength test was 29

Comments 9 • Page 4, Lines 186–188: The sentence “A questionnaire on pain …” is grammatically unclear and should be rephrased. 
Response 9: Thank you for pointing this out. We agree with this comment. We have changed all the text for the Moberg Pick up Test
Updated text in the manuscript in Page 4 Paragraph 6and line 189

Comments 10: Results While you were asked to provide results for the affected versus non-affected hand, Table 2 presents results for dominant and non-dominant hands (i.e. the results of Moberg and JTHFT). Please clarify this discrepancy. 
Response 10: Thank you for pointing this out. We agree with this comment. We have changed all the text for the Moberg Pick up Test
Updated text in the manuscript in Page 7 Table 2

Comments 11: Discussion 
I regret to inform you that the discussion section still does not adequately address my previous concerns. The primary issue remains a lack of focus on your own findings and overreliance on tangential topics (e.g., general rehabilitation strategies, psychological impacts, or unrelated studies). Your study has four key findings: 
• The effect of age on grip strength. 
Response  11: Thank you for pointing this out. We agree with this comment. We have changed the text.
Updated text in the manuscript in Page 10 last Paragraph and line 286-292

Comments 12• The relationship between time since surgery and pinch strength recovery. 
Response 12: Thank you for pointing this out. We agree with this comment. We have changed the text.
Updated text in the manuscript in Page 11 Paragraph and line 293-300

Comments 13• The association between employment status and disability (i.e., SPADI score). 
Response 13: Thank you for pointing this out. We agree with this comment. We have changed the text.
Updated text in the manuscript in Page 11 Paragraph and line 293-300
Employment

Comments 14• The predictors of shoulder pain and disability. 
I suggest the first paragraph of the discussion should briefly reiterate the study's purpose and summarize these key findings. Subsequent paragraphs should then be dedicated to each key finding, thoroughly exploring their implications and comparing them with relevant studies. Extraneous topics should be avoided. 
Response 14: Thank you for pointing this out. We agree with this comment. We have changed the text.
Updated text of First paragraph,   in the manuscript in Page 10 Paragraph and line 278-285
We update about The predictors of shoulder pain and disability. In page 11 and line 305-312

Comments 15: Furthermore, a separate paragraph should address the study’s limitations (e.g., cross-sectional design, small sample size, measurement tools, and unmeasured confounders such as rehabilitation protocols, psychosocial status, or comorbidities). 
Response 15: Thank you for pointing this out. We agree with this comment. We have changed the text.
Updated text in the manuscript in Page 12 Paragraph 4 and line 349-354

Comments 16: Finally, the discussion should conclude with the clinical implications of your study and suggestions for future research, grounded in your findings. For example, emphasize the importance of post-surgical monitoring of upper limb functionality, particularly grip strength and fine motor skills. 
Response 16: Thank you for pointing this out. We agree with this comment. We have changed the text.
Updated text in the manuscript in Page 12, Paragraph 5 and line 355-360

Comments 17:  I urge you to carefully revisit the Discussion section to ensure it directly interprets your results and avoids unsupported extrapolations. As it stands, the current version risks rejection due to its lack of focus. 
Response 17: Thank you for pointing this out. We agree with this comment. We have changed the text more or les 50% of them

Comments 18 Conclusion:  The final paragraph of the conclusion overlaps with the Discussion. Integrate its content into the Discussion’s clinical implications or limitations sections. 
Response 18: Thank you for pointing this out. We agree with this comment. We have changed the text.
Updated text in the manuscript in Page 12, Paragraph 5 and line 355-360

Comments 19- Line 342: “SPADI scores” should be “total SPADI score”. 
Response 19: Thank you for pointing this out. We agree with this comment. We have changed the text.
Updated text in the manuscript in Page 12. Paragraph 6 and line 368

Reviewer 2 Report

Comments and Suggestions for Authors

My first decision was rejection 

Author Response

No comments.

Reviewer 4 Report

Comments and Suggestions for Authors

The reviewer appreciates the authors' meticulous work and kindly requests that the title be formatted in sentence case.

Author Response

Comments: The reviewer appreciates the authors' meticulous work and kindly requests that the title be formatted in sentence case

Response: Thank you for pointing this out. We agree with this comment. We have changed the text.
Updated text in the title, in Page 1, line 2-4